# Polysaccharide Coating of Gelatin Gels for Controlled BSA Release

**DOI:** 10.3390/polym11040702

**Published:** 2019-04-17

**Authors:** Jimena S. Gonzalez, Carmen Mijangos, Rebeca Hernandez

**Affiliations:** 1Institute of Materials Science and Technology (INTEMA), University of Mar del Plata and National Research Council (CONICET), Colón 10890, 7600 Mar del Plata, Argentine; 2Instituto de Ciencia y Tecnología de Polímeros (ICTP), CSIC, Juan de la Cierva 3, Madrid, 28006 post code, Spain; cmijangos@ictp.csic.es (C.M.); rhernandez@ictp.csic.es (R.H.)

**Keywords:** LBL coating, multilayer film, gelatin gel, protein release

## Abstract

Self-assembly of natural polymers constitute a powerful route for the development of functional materials. In particular, layer-by-layer (LBL) assembly constitutes a versatile technique for the nanostructuration of biobased polymers into multilayer films. Gelatin has gained much attention for its abundance, biodegradability, and excellent gel-forming properties. However, gelatin gels melt at low temperature, thus limiting its practical application. With respect to the above considerations, here, we explored the potential application of gelatin gels as a matrix for protein delivery at physiological temperature. A model protein, bovine serum albumin (BSA), was encapsulated within gelatin gels and then coated with a different number of bilayers of alginate and chitosan (10, 25, 50) in order to modify the diffusion barrier. The coated gel samples were analyzed by means of Attenuated Total Reflectance-Fourier Transform Infrared (ATR-FTIR) and confocal Raman spectroscopy, and it was found that the multilayer coatings onto polymer film were interpenetrated to some extent within the gelatin. The obtained results inferred that the coating of gelatin gels with polysaccharide multilayer film increased the thermal stability of gelatin gels and modulated the BSA release. Finally, the influence of a number of bilayers onto the drug release mechanism was determined. The Ritger-Peppas model was found to be the most accurate to describe the diffusion mechanism.

## 1. Introduction

Polymers obtained from the biomass, known as natural polymers, are comprised of a broad family that can be divided into polysaccharides, proteins and lipids. Natural polymers, specifically polysaccharides and proteins, are inherently biodegradable, biocompatible, and non-toxic, characteristics that makes them good candidates for the development of biomaterials to be employed in a wide range of biomedical applications [1,2,3,4]. In addition, they also serve as novel packaging components improving transport properties and acting as active antibacterial and antiviral materials [5], or as environmentally safe adhesives [6]. Self-assembly of natural polymers constitute a powerful route for the development of functional materials in the form of nanoparticles, films or gels among others [7,8]. In particular, layer-by-layer (LBL) assembly that consists of the sequential adsorption of oppositely charged polyelectrolytes onto a substrate, constitute a versatile technique for the nanostructuration of biobased polymers into multilayer films. Such LBL multilayer films have been employed in a wide range of biomedical applications [9,10,11,12,13], among them, polysaccharide multilayer films have been employed as model surfaces for the study of the adhesion of living cells provided that the chemical and physical surface properties of the films were key to the modulation of cell adhesion [14,15,16,17]. Drug delivery is another important application of multilayer films due to the fact that the LBL technique allows the encapsulation of different therapeutic drugs into multilayer films by direct assembly during the build-up process [12,13,18,19,20]. Taking into account that this assembly is carried out in aqueous solutions under mild conditions, the activity of the loaded drugs was retained in the films [21]. Very recently, alginate and chitosan multilayer films have been employed as platforms for local sustained release of tamoxifen. Results demonstrated that sustained tamoxifen was released over time and that the release rate can be modulated with a number of deposited bilayers [22]. 

Gelatin is a protein derived from collagen that has gained much attention for its abundance, biodegradability, and excellent gel-forming properties. However, gelatin has low mechanical strength and dissolves relatively quickly in aqueous solution, in fact, as it is well known, gelatin gels melt at low temperature (around 35 °C), thus limiting its practical applications. Specifically, for drug delivery applications, chemical crosslinking of the gelatin gels is often employed as a strategy to maintain its structural stability at physiological temperature [23]. In addition, even though hydrogels based on gelatin have been reported for their protein delivery, protein drug loaded directly into gelatin hydrogels was found to release quickly due to the large pore size of the gel network [24]. In this regard, chemical cross-linked gelatin gels allow for sustained release of proteins, for example, lysozyme [25]. 

It is well known that LBL coating of hydrogels which are intended for drug release applications, allows them to modulate responsiveness and swelling properties and this strategy has been mainly employed for drug loaded nanocarriers [26,27,28,29,30]. In fact, LBL coating of gelatin nanoparticles has been reported before in order to control the loading/release characteristics of polyphenols and at the same time, modulate nanoparticle cell uptake rates and the ratio of gelatin nanoparticles within breast cancer cells [31]. In this present research, we intend to explore the potential application of gelatin hydrogels as a matrix for protein delivery at physiological temperature (above gelatin gel melting) in the absence of chemical crosslinking. To this aim, the coating of gelatin gels with polysaccharide multilayer films is proposed to obtain a fully biobased drug delivery matrix for protein delivery. To the best of our knowledge, this is the first time that layer-by-layer (LBL) polysaccharide coating has been applied onto macroscale gelatin hydrogels as a strategy to increase the thermal stability of gelatin gels and to modulate the release of a model protein drug. Furthermore, the effect of the interpenetration of the polyelectrolyte coating within the gel and its ability to control protein release has never been investigated. 

In this study, bovine serum albumin (BSA) was encapsulated within gelatin gels and sandwiched within a varying number of bilayers of alginate (Alg) and chitosan (Chi). Alginate/chitosan LBL coating was fabricated by means of a spray-assisted LBL assembly employing automatic equipment, and the interaction between the polyelectrolyte coating and the gelatin gel were analyzed by means of ATR-FTIR spectroscopy. Confocal Raman spectroscopy was employed as the main experimental probe to determine the degree of interpenetration of the polysaccharide coating within the gelatin gel. The results obtained were correlated with the release mechanism determined for BSA as a function of the number of bilayers.

## 2. Materials and Methods 

### 2.1. Materials

Gelatin from the porcine skin was provided by Sigma (G1890, lot SLBR2368V, Saint Louis, MO, USA). Low molecular weight chitosan (Chi) was supplied by Aldrich (448869, lot SLBG1673V, Saint Louis, MO, USA). Sodium alginate (Alg) was supplied by Sigma-Aldrich (A2158, lot 090M0092V, Saint Louis, MO, USA). Poly(ethylenimine) (PEI), with a molecular weight (Mw) of 25,000, and acetic acid were supplied by Aldrich and used as received. Sodium acetate anhydrous was supplied by Panreac and chloride acid by VWR. Albumin-fluorescein isothiocyanate conjugate (albumin bovine, BSA) was provided by Sigma (A9771, lot SLBP0519V) and phosphate buffer solution (PBS) of pH 7.4 was from Sigma-Aldrich (Lot SLBW0551).

### 2.2. LBL Coating of Gelatin Gels

The preparation of Alg/Chi multilayer film was carried out by automatic spray assisted LBL equipment (Nadetech Innovations, Navarra, Spain). Aqueous solutions of alginate, 2.5 mg/mL in aqueous buffer solution at pH = 3 (sodium acetate 0.1 M and acetic acid 0.1 M) and chitosan, 1 mg/mL in aqueous buffer solution at pH = 5 (sodium acetate 0.1 M and acetic acid 0.1 M), were alternatively sprayed onto glass slides (diameter of 14 mm) positively charged with a layer of polyethylenimine (1 mg/mL). The spray of the polymer solution was controlled by the compressed air (150 bar), the deposition time (5 s) and the drying time (15 s) after each polymer deposition (each layer used a volume of 0.85 mL approximately).

An amount of 400 µL of an aqueous gelatin solution (10% *w/v*) was deposited on top of an Alg/Chi multilayer film ending in alginate with a number of bilayers of n + 1. Figure 1 shows the methodology employed to select the experimental conditions employed. The sample was maintained at 5 °C for 1 h to allow for the physical gelation of gelatin. Finally, a multilayer Alg/Chi film (n bilayers) was built on top of the gelatin gels. Samples were stored at 5 °C for 24 h before testing. The samples were designated as SandG_n where n denotes the total number of bilayers (10, 25 and 50). 

For comparison, a blank gelatin gel (GelG) was prepared through deposition on 400 µL of an aqueous gelatin solution (10% *w/v*) onto a glass substrate. The sample was left at 5 °C during 24 h to allow for the physical gelation of gelatin. 

A series of precipitation studies were carried out in order to determine the best conditions for the interaction of gelatin with both sprayed polymers. This was detailed in Appendix A.

The thickness of the samples was measured with a Mahr Millitast 1085 Digital Indicator with a resolution of 1 µm.

### 2.3. Morphological, Characterization 

Scanning electron microscopy (SEM) was employed to characterize morphologies. SEM images were obtained using a SEM Philips XL30, instrument (Philips, Eindhoven, Netherlands). Samples were lyophilized, cryo-fractured by a previous immersion in liquid N2 and coated with gold before testing. 

### 2.4. Chemical Characterization

Fourier transform infrared (FTIR) spectroscopy measurements were done in a Perkin Elmer UATR TWO spectrometer (PerkinElmer limited, Seer Greeb, UK), employing a resolution of 4 cm^−1^. Measurements were carried out in attenuated total reflectance modes (with ATR accessory) from 400 to 4000 cm^−1^, performing 16 scans per sample over the surface in a dried state.

Raman spectra were taken in a Renishaw InVia Reflex Raman system (Renishaw plc, Old Town, Wotton-under-Edge, UK). It was used by employing a grating spectrometer with a Peltier-cooled charge-coupled device (CCD) detector, coupled to a confocal microscope. All spectra were processed using Renishaw WiRE 3.3 software. The scans were obtained from the surface of the samples and the depth profiles were carried out positioning the laser each 2 μm.

### 2.5. Drug Release Experiments

Bovine Albumin (BSA) was used as a model protein to investigate the controlled release of the coated gels. One milligram of BSA for each 100 mg of gelatin was added in the gelatin solution before multilayer coating. The dried samples were cut (0.5 × 0.5 cm, approximately 10 mg) and immersed in 3 mL of phosphate buffer solution (PBS) of pH 7.4 (Sigma-Aldrich) at 37 °C.

The concentration of BSA was monitored at 495 nm using the UV–Vis spectrophotometer, Nanodrop One (Thermo Scientific, Waltham, MA, USA. A volume of 100 μL of release medium was sampled with replacement. The cumulative amount of BSA released as a function of time was calculated. Each experiment was performed three times, and the mean values and standard deviations were calculated.

## 3. Results and Discussion

### 3.1. Physicochemical Characterization of LBL Coated Gelatin Gels (SandG_n)

Table 1 shows the thickness of all the samples under study. As can be observed, all samples presented sample thickness of ~100 μm except for the sample SandG_50 for which thickness was slightly increased to 125 ± 12 µm.

Figure 2a,b showed representative photographs corresponding to the sample SandG_25 where it can be observed that the coated LBL gelatin films retained the flexibility of the pristine gelatin film. The microstructure of the coated films was examined using a scanning electron microscopy (SEM) after lyophilization of the samples. Figure 2a,b shows representative photographs corresponding to the sample SandG_25. As it can be observed, the sample can be detached from the glass substrate where it was fabricated (Figure 2a) without any post-processing step. In addition, it can be easily handled with tweezers and retained the flexibility of the pristine gelatin film (Figure 2b). The morphology of the sample SandG_25 was also observed in cross section (Figure 2c), revealing a porous microstructure throughout the sample with a heterogeneous pore size distribution. Such morphology is characteristic of lyophilized gelatin gels [23]. 

The analysis of chemical structure by ATR-FTIR spectroscopy allowed us to ascertain, on the one hand, that the alginate and chitosan were effectively incorporated onto the gelatin gel and, on the other hand, the presence of physical interactions in-between the gelatin and the polyelectrolyte coating. Figure 3 shows the ATR-FTIR spectra in the region 1800–1000 cm^−1^ taken on the surface of LBL gelatin gels coated with different numbers of Alg/Chi layers (n = 10, 25, 50). The ATR-FTIR spectra corresponding to blank gelatin gel, chitosan and alginate (as received) was also shown for comparison.

Sodium alginate (as received, Figure 3, Alg) displayed two absorption bands assigned to the carboxylate group, an antisymmetric stretch at 1596 cm^−1^ and a symmetric stretch at 1412 cm^−1^. Chitosan (as received, Figure 3, Chi) presented bands at 1647 cm^-1^ assigned to amide I and 1586 cm^−1^ assigned to N−H bending from amine and amide II. ATR-FTIR spectra of alginate and chitosan displayed a strong band at 1026 cm^−1^ corresponding to skeletal vibration of C–O stretching of the polysaccharidic backbone [32]. The ATR-FTIR spectra corresponding to the gelatin gel (Figure 3, GelG) presented two strong amide I and amide II bands at 1630 and 1543 cm^−1^, respectively [33,34].

As it is clearly observed, in the spectra corresponding to LBL gels there was an increase of the relative intensity assigned to the band located at 1026 cm^−1^, corresponding to the skeletal vibration of C-O stretching of the polysaccharidic backbone of alginate and chitosan, with respect to that located at 1630 cm-1 assigned to amide I of gelatin. This result suggests the deposition of an increased amount of polysaccharides onto the gelatin gel with the number of Alg/Chi bilayers [35].

Determination of the presence of ionic interactions in-between the gelatin and the polyelectrolyte coating was challenging due to the fact that the carbonyl vibrations of the carboxylate and carboxylic acid group (responsible for ionic interactions in the alginate), the N−H vibrations of amines and protonated amines (responsible for ionic interactions in chitosan) and N–H vibrations of the amides in gelatin overlapped in the region 1700–1450 cm^−1^. For the same reason, these are not conclusive results about the formation of ionic interactions between the alginate and chitosan in LBL films as determined through FTIR [32]. Some information can be extracted from a comparison of the ATR-FTIR spectra depicted in Figure 3. The band located at 1647 cm^−1^ shifted to lower wave numbers with the number of Alg/Chi bilayers deposited onto the gelatin gel with respect to the ATR-FTIR spectra corresponding to the gelatin gel. In addition, the intensity corresponding to the band located at 1543 cm^−1^ increased with respect to that located at 1647 cm^−1^. These results might suggest the establishment of ionic interactions involving positively charged amide groups from the gelatin.

### 3.2. Determination of the Degree of Interpenetration within LBL Coated Celatin Cels 

As it has been shown in the literature, alginate and chitosan multilayer films build up through sequential deposition of their aqueous solutions and present some degree of interpenetration which can be attributed to the diffusion of chitosan “in” and “out” of the film during the deposition process [36]. Likewise, it has been reported that multilayer coatings onto polymer micro particles can be interpenetrated to some extent within the polymer matrix [27,33].

In order to quantify the degree of interpenetration of the LBL coating within the gelatin gel, confocal Raman spectroscopy was carried out on dried samples as a function of depth to determine changes in chemical composition that correspond to the LBL polyelectrolyte coating of the gelatin gel with alginate and chitosan. Figure 4 shows the depth profile of normalized Raman spectra corresponding to the sample SandG_25. For comparison, the Raman spectra corresponding to GelG and that corresponding to alginate that constitutes the last deposited layer were also depicted in Figure 5. Depth profiles of normalized Raman spectra corresponding to the sample SandG_10 and SandG_50 were shown in Appendix A.

The Raman spectrum corresponding to GelG shows amide I (~1667 cm^−1^), CH2 scissoring (~1450 cm^−1^) and amide III (~1245 cm^−1^) bands [24,35]. A sharp aromatic ring breathing peak (~998 cm^−1^) was clearly visible in the spectrum corresponding to GelG which probably arose from phenylalanine amino acid residues [37]. The Raman spectra corresponding to SandG_25 (0 μm) shows the appearance of a shoulder located at 1640 cm^−1^ and an increase in the relative intensity of the band located at 1409 cm^−1^ with respect to that located at 1460 cm^−1^ when compared to the spectra corresponding to the gelatin gel. The band at 1460 cm^−1^ could be attributed to the appearance of the carboxyl ion stretch (typical for alginic acid salts) as a consequence of the partial protonation of Na-Alginate at the experimental conditions employed for the preparation of the LBL gels. This is in agreement with an increase in the relative intensity of the band located at 1409 cm^−1^ in the spectra corresponding to SandG_25 which could be attributed to symmetric carboxyl stretching of alginic acid [38].

Figure 5 depicts the ratio between the intensities of the Raman peaks located at 1409 cm^−1^ and 998 cm^−1^ (I_1409_/I_998_) for SandG samples as a function of the depth starting from the surface up to 8 μm within the sample. The value corresponding to GelG was also added in the plot for comparison. As can be observed, SandG_25 and SanG_50 samples showed a drastic decrease in the intensity ratio (I_1409_/I_998_) up to ~2 µm depth. From this value up to 8 µm, the intensity ratio (I_1409_/I_998_) for SandG_50 reached a plateau whereas the intensity ratio (I_1409_/I_998_) corresponded to the sample SandG_25 continued to decrease and the values obtained tended to be those exhibited by GelG. For the sample SandG_10 the intensity ratio (I_1409_/I_998_) did not change with the depth. Recent results published on LBL Alg/Chi films prepared under the same experimental conditions employing an automatic LBL coater have revealed that the thickness corresponded to Alg/Chi films with 25 bilayers was 350 nm [22]. Therefore, from the results obtained, this suggested that there was a high degree of interpenetration of the polysaccharide coating within the gelatin gel because of the presence of polysaccharide chains entangled within the gelatin gel which was much more evident within the first three microns of the sample for the samples SandG_25 and SandG_50. On the contrary, SandG_10 sample seemed to be whole fully interpenetrated because no drastic changes in the intensity ratio were shown as a function of depth. A schematic representation of the structural organization of the LBL coated gelatin gels is shown as an inset of Figure 5. 

### 3.3. Protein Release Behavior 

Figure 6 shows representative pictures of pristine gelatin (GelG) and sample SandG_25 after being immersed in water for two hours at room temperature and at 37 °C. As can be observed, GelG was completely dissolved after being immersed in water for two hours at 37 °C whereas sample SandG_25 remained stable when maintained in water at 37 °C for two hours. These results confirmed that the polysaccharide coating gave rise to an increased thermal stability of the gelatin hydrogel. These findings were related to BSA release from GelG and SandG samples through measurements of BSA cumulated release at 37 °C carried out as a function of time (Figure 7).

For GelG, it was shown that the release of BSA was accelerated with the gradual liquidity of the gel. While less than 10% of the BSA was released within the first 15 min, a 100% release of BSA was achieved after 45 min which was the time for the gel-sol phase change of the gelatin hydrogel. In contrast, SandG samples showed a more sustained release dependent on the number of bilayers of the Alg/Chi coating, so that after 45 min SandG_10 released 50% of the BSA, SandG_25 released 30% of the BSA and SandG_50 released less than 30% of the BSA. 

Different from sample GelG that immediately melted after the initial burst release, SandG samples showed a continuous release until completely melting and presented biphasic release profiles. In the first step there was a burst release followed by a plateau and a second step characterized by continuous release until complete melting of the SandG samples. Therefore, for SandG samples, the polysaccharide coating, in addition to increasing the thermal stability of the gelatin gels, might act as a diffusion barrier to retard BSA release from the gelatin gel. Furthermore, the polysaccharide coating could block the surface macropores of the gelatin, leading to reduced water penetration through the slower pores and slower drug release rate [39].

In order to deepen into the mechanism of drug delivery, the cumulative drug release of up to 0.4 (corresponding to the first release step, as shown in Figure 7) was fitted to the Ritger-Peppas model for samples SandG_25 and SandG_50 (Figure 8). It is important to note that neither the sample GelG nor the sample SandG_10 could be fitted to this model. The fact that samples GelG and SandG_10 could not be fitted to diffusion controlled models was in agreement with similar release profiles encountered for these two samples as shown in Figure 7. This result could be related to the high degree of interpenetration in between the polyelectrolyte layers and gelatin as demonstrated through Confocal Raman spectroscopy.

The Ritger-Peppas model is described by Equation (1) [40]: (1)$$\frac{M_{t}}{M_{\infty }}=k\cdot t^{c}$$
where *M*_t_/*M*_∞_ is the released fraction and *k* and c are fitting parameters characteristic of the film/solution medium and transport mechanism, respectively. The value of c indicates Fickian diffusion (c = 0.5), anomalous transport (0.5 < c < 1) or Case-II transport (c = 1). 

The fitting parameters obtained from the Ritger-Peppas model are summarized in Table 2. The c value obtained for SandG_25 was 0.57, which confirmed the diffusion-controlled drug release mechanism that suggests a Fickian transport mechanism (expected value around n = 0.5). As to the sample SandG_50, the n value = 0.68 indicates an anomalous non-Fickian mechanism (mixed diffusion and relaxation mechanism). The transition from the Fickian to the non-Fickian mechanism might be attributed to the presence of the polyelectrolyte layers added without interpenetration with the gelatin as demonstrated by the confocal Raman spectroscopy.

## 4. Conclusions

A novel coated hydrogel was obtained and the final properties of the material were modulated by the numbers of bilayers. The multilayers Alg/Chi were interpenetrated with gelatin gel. Results showed sustained BSA release over time and that the release rate can be modulated with the number of deposited bilayers. The release mechanism was diffusion controlled using the Ritger-Peppas model as the most accurate to describe the diffusion mechanism process. The strategy reported herein opens new routes for the development of fully biodegradable release patches based on gelatin.

## Figures and Tables

**Figure 1 polymers-11-00702-f001:**
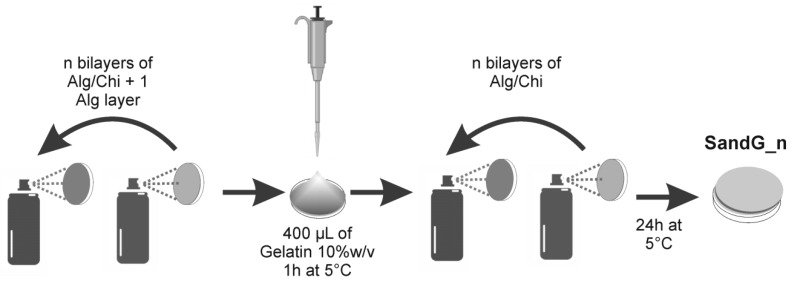
Schema of multilayered coating of gelatin gels.

**Figure 2 polymers-11-00702-f002:**
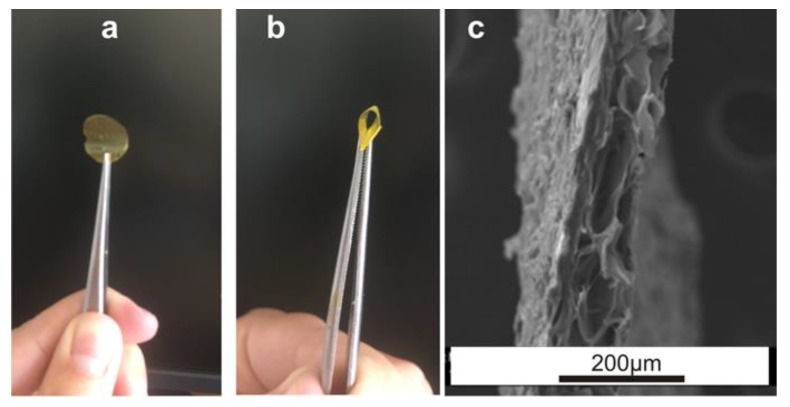
(**a**,**b**) Photographs of SandG_25. (**c**) SEM image corresponding to the cross sections of SandG_25.

**Figure 3 polymers-11-00702-f003:**
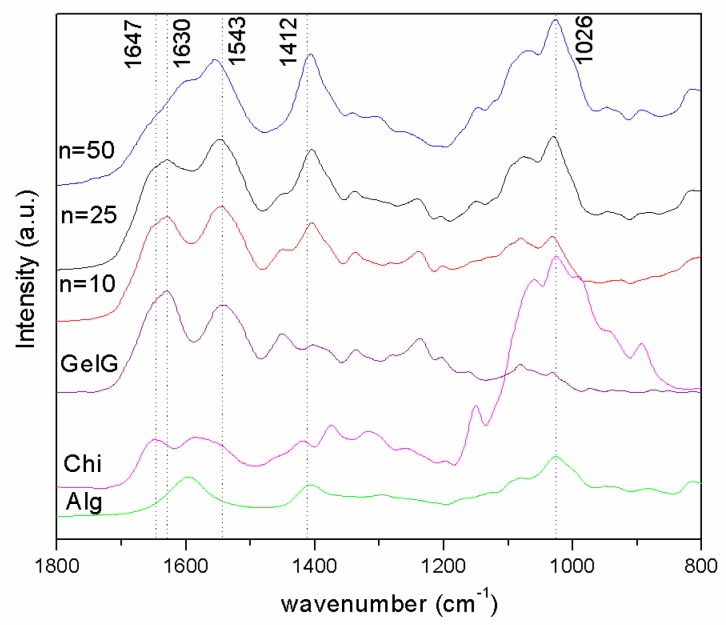
Attenuated Total Reflectance-Fourier Transform Infrared (ATR-FTIR) spectra of alginate and chitosan (as received), GelG and SandG samples (n = 10, 25 and 50).

**Figure 4 polymers-11-00702-f004:**
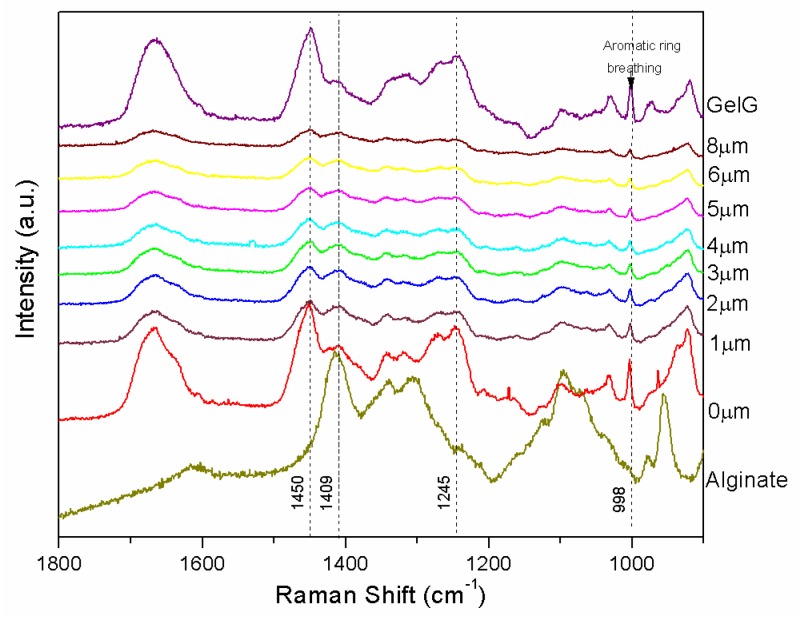
Depth profile of normalized Raman spectra corresponding to the sample SandG_25. For comparison, the spectra corresponding to GelG and alginate are also depicted.

**Figure 5 polymers-11-00702-f005:**
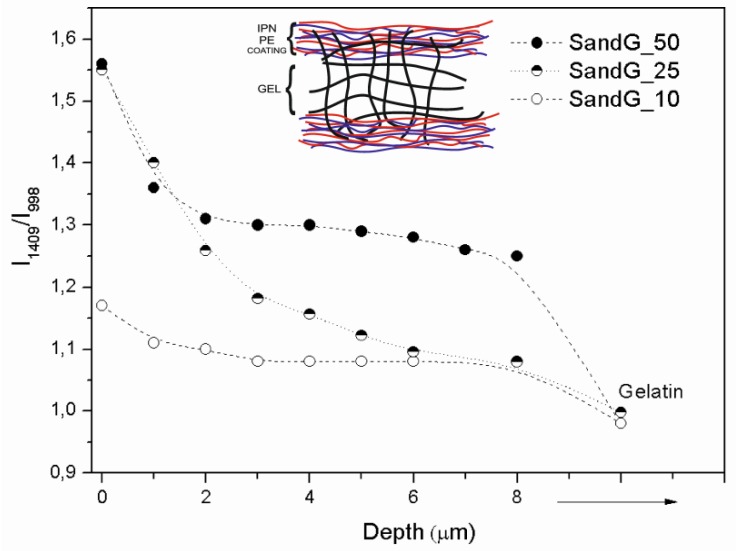
Peak intensity ratios vs. depth for SandG_10, SandG_25 and Sand_50.

**Figure 6 polymers-11-00702-f006:**
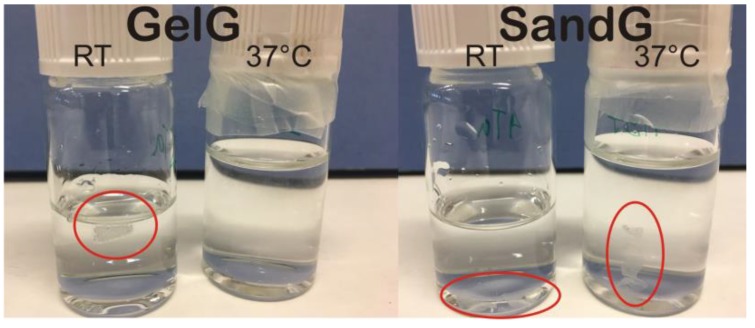
Dissolutions studies at room temperature and at 37 °C of samples GelG and SandG_25 after being immersed for 2 h.

**Figure 7 polymers-11-00702-f007:**
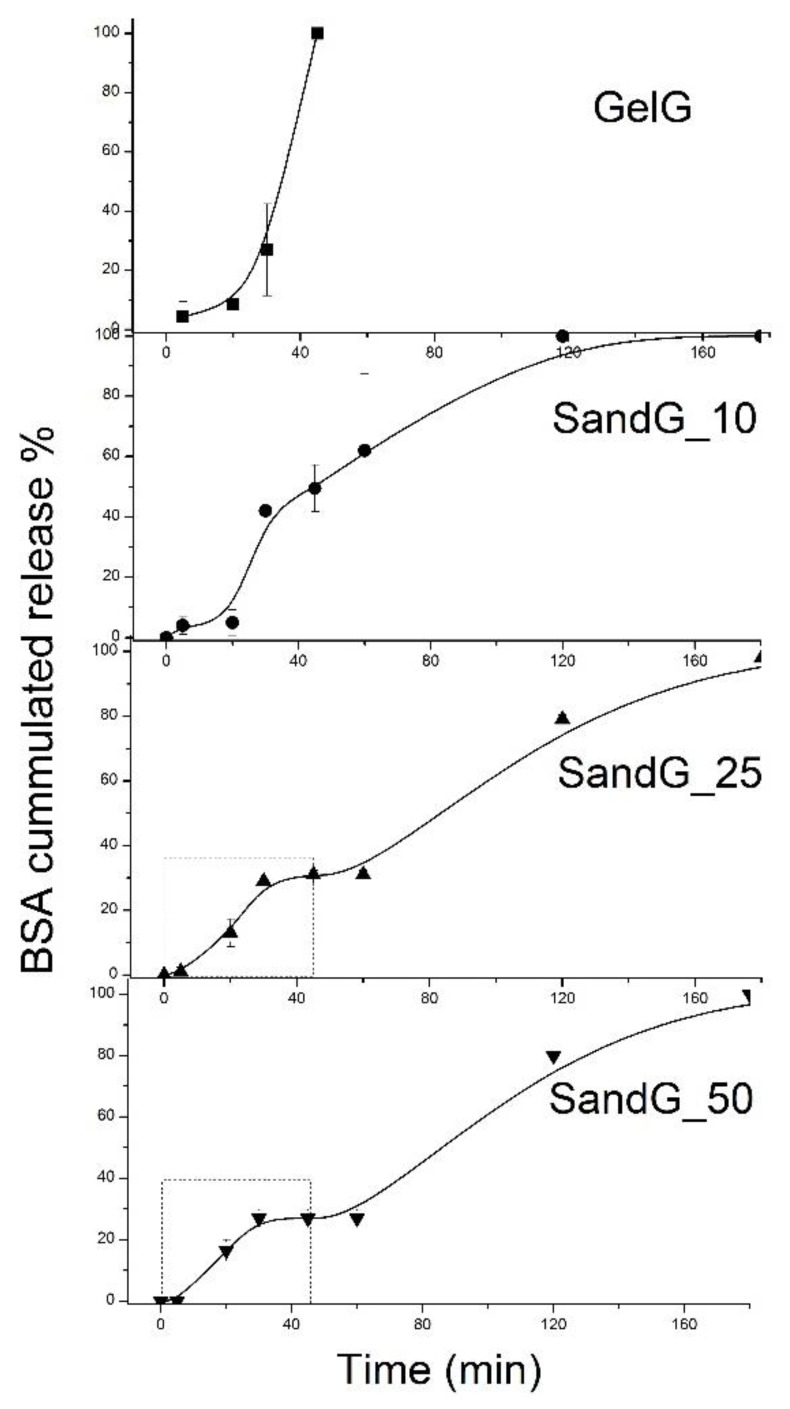
Cumulative time-dependent release profiles of BSA from all the samples under study: (**a**) GelG; (**b**) SandG_10; (**c**) SandG_25 and (**d**) SandG_50. Dashed lines distinguished different steps within the release profile for all samples. The squares depicted in the plots corresponding to SandG_50 and SandG_25 marked the region employed for the curve fitting shown in Figure 8.

**Figure 8 polymers-11-00702-f008:**
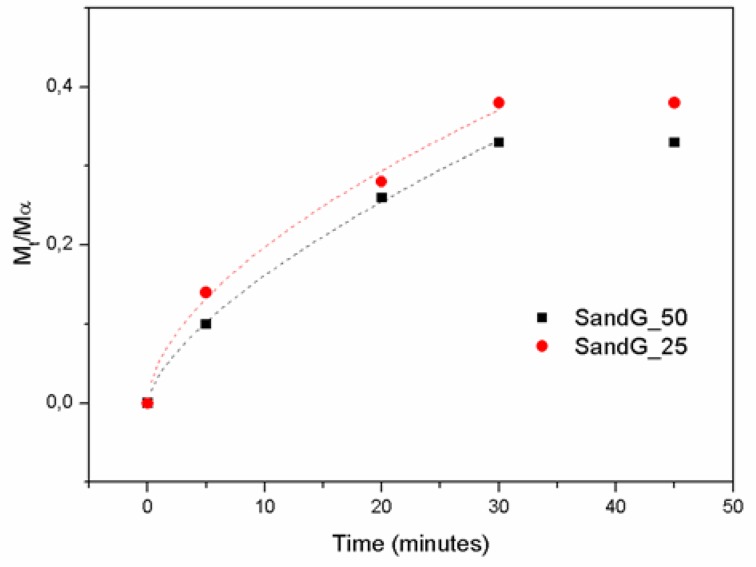
Cumulative time-dependent release profiles of BSA (Bovine Serum Albumin) from multilayer films. The fitting with dashed lines confirmed to the Ritger-Peppas model up to 0.4 cumulative BSA release.

**Table 1 polymers-11-00702-t001:** Thickness of all the samples under study.

Sample Designation	N Bilayers	Thickness (µm)
GelG	0	97 ± 2
SandG_10	10	98 ± 18
SandG_25	25	104 ± 12
SandG_50	50	125 ± 12

**Table 2 polymers-11-00702-t002:** Parameters obtained through fitting to the Ritger-Peppas model.

Sample	k	c	R2
**SandG_25**	0.0528 ± 0.0132	0.5735 ± 0.0793	0.9773
**SandG_50**	0.0357 ± 0.0039	0.6561 ± 0.0343	0.9971

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
