# Peer review of "Polysaccharide Coating of Gelatin Gels for Controlled BSA Release"

_polymers, 2019, doi:10.3390/polym11040702_

Reviewer 1 Report

I suggest publication of the manuscript in its present form.

Reviewer 2 Report

As compared to last version of manuscript, the reviewer thinks that the authors have extensively revised the organization of the manuscript, which make the paper read smoothly. The authors arranged the order of figures and text in the results and discussion. Now the quality of the manuscript has been greatly enhanced. Other than these good points, the authors should change the title of the supplementary materials. 

This manuscript is a resubmission of an earlier submission. The following is a list of the peer review reports and author responses from that submission.

Round  1

Reviewer 1 Report

The manuscript by Gonzalez et al. described the fabrication of coated gelatin gels for release of bovine serum albumin. Although the manuscript fits in the scope of the Journal, the design of the experiment is not well thought. As such, it is not recommended for publication.

(1)Line 131, there is a typo for the unit of the thickness (i.e., 100 mm)

(2)Figure 3 should not be called swelling assay. It is a simple dissolution study. Please change the caption of the figure.

(3)What is the meaning of using FTIR to investigate the LbL samples and comparing to the blank gelatin gel, chitosan and alginate? The technique that the authors used is by physically assembling the layers. Are the authors suggesting potential bonding interactions from the physical assembly of the LbL?

(4)A coated gel that does controlled release has been extensively studied, what is the scientific merit of the manuscript?

Reviewer 2 Report

This paper prepares lbl materials consisting of gelatin, chitosan and alginate to evaluate the drug releasing effectiveness of these lbl materials encapsulating BSA. The authors use spin-coating method to prepare lbl materials and use the Ritger-Peppas model to evaluate the diffusion mechanism of BSA through lbl materials. This study presents a few major issues that may require an intensive revision. Some comments and suggestions are given below:

1. The reviewer is confused with the word “nanocoating”. The authors do not focus on the nanocoating or saying, the authors do not often use this word in the manuscript. It is only present twice in the text. What does “nanocoating” mean? The thickness of lbl materials that the authors prepare is microscale or even bigger. What else in this study can be used to represent nanocoating? Please specify.  

2. If “nanocoating” is important, the authors should add background information in the introduction about what the nanocoating technology is and how other studies make use of this technology.

3. The method of LbL coating of gelatin gels needs to be more specific. Chitosan solution and alginate solution are alternatively sprayed on the glass slides. The number of bilayer is counted as 10, 25, and 50. It not clear that when to end the last spray and when to start spraying the next solution. What is the metric to exchange the solution?

4. Figure 2c needs to be more specific to label the individual layers. What is the bilayer and what is the gelatin containing BSA?

5. The narrative from Line 134 to Line 143 is not clear. Particularly, staring from Line 139, the authors talk about previous SEM studies, while there is no reference provided and how these two study are related. The authors need to clarify.

6. Some important peaks in Figure 4 and Figure 5 need to be labeled in the figures.

7. The reviewer is confused with the FTIR and Raman. If the authors choose the sample surface to scan rather than the cross-section, it is hard to find the difference amongst all the samples, because almost all the samples will have similar surface consisting of bilayers of chitosan and alginate except the control sample. If the authors choose the cross-section of the samples to scan, this should be clarified in the method section and in the results and discussion.

8.  The English writing is not reliable and sometime is hard to follow. Many typos, grammatical errors, and long sentences are present. The paper organization is not good as well. The results and discussion section may require a re-written. It is hard to find the connection between one characterization and another one.

Reviewer 3 Report

Gonzalez et al. report results on the construction and structural characterization of novel hierarchical structures of gelatin gels coated with alginate/chitosan nano-multilayers, using the LbL technique. The authors also demonstrated the application of such nanostructures for the delivery of BSA as a model protein. The idea is original. Several novel insights are introduced while discussions are comprehensive.

I have some minor comments:

1. Fig. 1: please indicate on the scheme the temperature of the gelatin application step as discussed in the text.

2. In relation to discussion in p. 3, l. 120-122: Can higher amounts of BSA be loaded in the nanocoated gels? What are the release profiles in these cases?

3. Fig. 5: change right axis legend to µm.

4. Discussion on FTIR spectra analysis: give some details on the normalization methodology used.

5. The BSA release profiles should be discussed in more details in respect to the different steps of the process shown in figure 7.